# Validation of Knock-Out Caco-2 TC7 Cells as Models of Enterocytes of Patients with Familial Genetic Hypobetalipoproteinemias

**DOI:** 10.3390/nu15030505

**Published:** 2023-01-18

**Authors:** Claire Bordat, Donato Vairo, Charlotte Cuerq, Charlotte Halimi, Franck Peiretti, Armelle Penhoat, Aurélie Vieille-Marchiset, Teresa Gonzalez, Marie-Caroline Michalski, Marion Nowicki, Noël Peretti, Emmanuelle Reboul

**Affiliations:** 1Aix-Marseille Université, INRAE, INSERM, C2VN, 13885 Marseille, France; 2CarMeN Laboratory, INSERM U1060, INRAE, UMR 1397, Université Claude Bernard Lyon 1, 69495 Pierre-Benite, France; 3Biochemistry Department, Hospices Civils de Lyon, 69495 Pierre-Benite, France; 4Pediatric Hepato-Gastroenterology and Nutrition Unit, Hôpital Femme Mère Enfant (HFME) de Lyon, Hospices Civils de Lyon, 69677 Bron, France

**Keywords:** abetalipoproteinemia, bioavailability, CRISPR/Cas9, chylomicrons, chylomicron retention disease, familial hypobetalipoproteinemia, lipoproteins, *PLIN2*, vitamin E, tocopherol

## Abstract

Abetalipoproteinemia (FHBL-SD1) and chylomicron retention disease (FHBL-SD3) are rare recessive disorders of lipoprotein metabolism due to mutations in *MTTP* and *SAR1B* genes, respectively, which lead to defective chylomicron formation and secretion. This results in lipid and fat-soluble vitamin malabsorption, which induces severe neuro-ophthalmic complications. Currently, treatment combines a low-fat diet with high-dose vitamin A and E supplementation but still fails in normalizing serum vitamin E levels and providing complete ophthalmic protection. To explore these persistent complications, we developed two knock-out cell models of FHBL-SD1 and FHBL-SD3 using the CRISPR/Cas9 technique in Caco-2/TC7 cells. DNA sequencing, RNA quantification and Western blotting confirmed the introduction of mutations with protein knock-out in four clones associated with i) impaired lipid droplet formation and ii) defective triglyceride (−57.0 ± 2.6% to −83.9 ± 1.6%) and cholesterol (−35.3 ± 4.4% to −60.6 ± 3.5%) secretion. A significant decrease in α-tocopherol secretion was also observed in these clones (−41.5 ± 3.7% to −97.2 ± 2.8%), even with the pharmaceutical forms of vitamin E: tocopherol-acetate and tocofersolan (α-tocopheryl polyethylene glycol succinate 1000). *MTTP* silencing led to a more severe phenotype than *SAR1B* silencing, which is consistent with clinical observations. Our cellular models thus provide an efficient tool to experiment with therapeutic strategies and will allow progress in understanding the mechanisms involved in lipid metabolism.

## 1. Introduction

Familial hypobetalipoproteinemias- (FHBL) related disorders, or Monogenic HypoBetalipoproteinemia Disorders, are classified into two pathophysiological groups: class I lipoprotein assembly and secretion defect (SD) and class 2 enhanced lipoprotein catabolism (EC) [1]. Class I includes intestinal genetic hypocholesterolemia that represents a heterogeneous group of inherited lipoprotein problems characterized by low plasma levels of triacylglycerol (TAG), total cholesterol, very-low-density lipoprotein (VLDL), low-density lipoprotein (LDL) and an almost complete absence of apolipoprotein B-100 and B-48. Intestinal recessive FHBL includes abetalipoproteinemia, previously called Bassen-Kornzweig syndrome (ABL or FHBL-SD1, OMIM 200100), familial hypobetalipoproteinemia owing to lipoprotein assembly defect 2 (FHBL or FHBL-SD2, OMIM 615558) and chylomicron retention disease or Anderson’s disease (CRD or FHBL-SD3, OMIM 246700) [2,3]. In this work, we focused on abetalipoproteinemia (FHBL-SD1) and chylomicron retention disease (FHBL-SD3). FHBL-SD1 is the consequence of mutations in the Microsomal Triglyceride Transfer Protein (*MTTP*) gene encoding for a protein (MTP) required for the assembly and secretion of apoB-containing lipoproteins: VLDL and chylomicrons from the liver or intestine [4]. Similarly, FHBL-SD3 is due to mutations in the *SAR1B* gene that are translated into the Secretion-Associated Ras-related GTPase 1B (Sar1b). Sar1b is involved in the initiation of chylomicron intracellular transport via COPII- (coat protein) coated vesicles, which is an essential step before the chylomicron secretion [5,6].

The absence of MTP and Sar1b activity causes the accumulation of large lipid droplets in enterocytes and hepatocytes, leading to the characteristic lipid profile described above with severe clinical consequences. Indeed, the FHBL-SD1 and FHBL-SD3 clinical phenotypes usually consist of intestinal lipid malabsorption, steatorrhea, fat-soluble vitamin (A, D, E, K) deficiency and chronic diarrhea leading to growth retardation during the neonatal period. These phenotypes are worsened by neurological and ophthalmic deterioration during the second decade of life, mainly related to vitamin A and E deficiencies [2]. Fat-soluble vitamin supplementation has to be started in early childhood together with dietary fat restriction to prevent such disabling complications [7]. However, despite early and adapted supplementation, previous studies showed that, unlike vitamins A, D and K, plasma levels of vitamin E are almost never normalized and remain chronically low [8], reaching only one-half to two-thirds of normal levels (20–40 μmol/L) [9] despite high doses of oral α-tocopherol (50–200 IU/kg/day).

Vitamin E, as a fat-soluble vitamin, follows the intestinal absorption, hepatic metabolism, and cellular uptake processes of other lipophilic molecules. Vitamin E absorption thus depends on chylomicrons formation and secretion [7]. Therefore, in the serum of untreated FHBL-SD1 and FHBL-SD3 patients, vitamin E is undetectable [9]. This affects the patients’ neurologic system causing complications ranging from peripheral neuropathy to cerebellar ataxia, as well as pigmentary retinitis. These neuro-ophthalmological complications become irreversible without appropriate vitamin E supplementation [10]. Anatomically, deep vitamin E deficiency is characterized by axonopathy with a swollen appearance and reduced myelination, as well as neuromuscular lesions with high levels of oxidative stress in neurons [11]. Functionally, affected neurons display attenuated axonal transport and altered mitochondrial respiratory control [12]. Thus, because vitamin E has a crucial role in the development and maintenance of the structural integrity of the nervous system, vitamin E deficiency is a major concern in patients with primary FHBL [9]. The mechanisms hampering vitamin E plasma level restoration in treated FHBL-SD1 and FHBL-SD3 patients are still unknown. We previously studied the mechanisms of absorption/secretion of α-tocopheryl polyethylene glycol succinate 1000 (TPGS, i.e., tocofersolan), a water-soluble derivative of α-tocopherol, compared to tocopherol acetate (TAC), the form usually used in clinical nutrition [13] by human wild-type Caco-2 TC7 cells, known to be a relevant model of the intestinal barrier [14]. In this work, we created a relevant knock-out (KO) cellular model of FHBL-SD1 and FHBL-SD3 by invalidating either *MTTP* or *SAR1B* in Caco-2/TC7 cells using the CRISPR/Cas9 genome-editing system. Our aim was to create and validate an in vitro model of enterocytes mimicking the enterocytes of FHBL-SD1 and FHBL-SD3 patients to explore lipid and vitamin E absorption in the context of these diseases.

## 2. Materials and Methods

### 2.1. Supplies

Dulbecco’s modified Eagle’s medium (DMEM) containing 4.5 g/L glucose GlutaMAX™, trypsin-EDTA, non-essential amino acids, penicillin/streptomycin and PBS were purchased from Life Technologies (Illkirch, France). Fetal bovine serum (FBS ref.: S1810-500) came from Dominique Dutscher SAS (Bernolsheim, France). DNA Primers, tracrRNA, crRNA, TrueCut™ Cas9 Protein v2 (Invitrogen™), Lipofectamine™ CRISPRMAX™ Cas9 Transfection Reagent (Invitrogen™), PureLink™ Genomic DNA kit (Invitrogen™), AmpliTaq Gold^®^ 360 DNA Polymerase (Applied Biosystems) and TRIzol reagent were from Thermo Fisher Scientific (Illkirch-Graffenstaden, France). Guide-RNA (gRNA) was designed by Invitrogen™ and produced by Synthego (Synthego Corporation, Redwood City, CA, USA). Sequences received at the laboratory were the following ones (for DNA sequences targeted by gRNA, see Appendix A Appendix A):*MTTP*-gRNA1: 5′-ACGCUCCUUCAUCUAAUCCA-3′*MTTP*-gRNA2: 5′-UACACGGCCAUUCCCAUUGU-3′*SAR1B*-gRNA1: 5′-UUGACUCUAACAGCCUUUCG-3′*SAR1B*-gRNA2: 5′-UCCGAAGAACUGACCAUGC-3′*HPRT1*-gRNA: 5′-GCAUUUCUCAGUCCUAAACA-3′ (positive control)

RRR-α-tocopherol, α-tocopherol acetate (TAC) and γ-tocopherol (used as an internal standard for vitamin E HPLC analyses) were from Sigma-Aldrich (Sigma-Aldrich Chimie by Merck, Saint-Quentin-Fallavier, France). α-tocopheryl polyethylene glycol succinate 1000 (TPGS) was contained in VEDROP^®^ 50 mg/mL (Orphan Europe, Puteaux, France). All solvents used were HPLC-grade from Carlo Erba Reagents (Peypin, France). Western-blot analyses were performed using either primary mouse monoclonal antibody raised against MTP (C-1, sc-515742), Sar1b (AT1C7, sc-517425) or β-actin (ACTBD11B7, sc-81178) from Santa Cruz Biotechnology INC., Dallas, TX, USA. Anti-mouse secondary antibody was an HRP-conjugated goat anti-mouse IgG antibody (goat, cat. No. 62-6620, 1:10,000; Zymed, Thermo Fisher Scientific).

### 2.2. Cell Culture

The Caco-2/TC7 cell line, derived from a human colon adenocarcinoma, naturally differentiates into a cell monolayer expressing several morphological and functional characteristics of enterocytes [15]. Caco-2/TC7 cells were cultured on 25 cm^2^ flasks in DMEM supplemented with 16% FBS, 1% of non-essential amino acids and 1% penicillin/streptomycin [13]. The percentage of FBS in the complete medium was then reduced from 16% (usual percentage) to 8% to ensure a correct growth of the KO clones. Cells were grown in a 37 °C humidified atmosphere with 90% air and 10% CO_2_.

For experiments, cells were split with 0.05% Trypsin-EDTA when 80% of confluence was reached and were seeded on permeable Millicell^®^ hanging cell culture filter inserts at a density of 3 × 10^4^ cells/cm^2^ (24 mm insert, 1 µm pore size PET membrane, Millicell^®^; Merck KGaA, Darmstadt, Germany) in 6-well plates, to reproduce the intestinal barrier. Cells were grown to confluence in a complete medium for one week. Cells were then cultured in asymmetric conditions, with 2 mL 8% FBS complete medium in the lower compartment and 1.5 mL serum-free complete medium in the upper compartment until total differentiation (around 21 days after seeding). The medium was changed every 2–3 days.

### 2.3. Genomic Primer Verification

To confirm the specificity of the genomic primers produced by Invitrogen™, a conventional genomic PCR protocol was followed. Commercial DNA was used to perform this assay, and 2 µL of it was mixed with 25 µL AmpliTaq^®^ Gold 360 Master Mix (Thermo Fisher Scientific) in the presence of 100 ng of the corresponding forward and reverse primers (see Appendix A Appendix A). The resulting PCR product (2 µL) was mixed with 8 µL of RNAse-free water and 2 µL of glycerol (30:100, *v*:*v*) and analyzed with a 1% E-gel EX agarose gel using BET as DNA intercalator. For genomic primers sequences used in CRISPR/Cas9 experiment, see Appendix A Appendix A.

### 2.4. Optimal Confluency Test

Cells were grown in 24-well cell culture plates to reach different confluency (from 30% to 50%) in each well to establish the optimal conditions for transfection. The cells were then transfected by the CRISPR/Cas9 gene editing system using Lipofectamine CRISPRMAX^®^ according to the protocol proposed by Invitrogen™. This test was conducted using a gRNA designed by Invitrogen™ to target the HPRT1 housekeeping gene considered a positive control. Cells were transfected as described below.

### 2.5. Generation of MTTP or SAR1B-KO Caco-2/TC7 Cells

Undifferentiated Caco-2/TC7 cells were seeded in 24-well cell culture plates at a density of 3 × 10^4^ cells/cm^2^. When they reached 30% confluence, cells were transfected with the TrueCut™ Cas9 V2 Proteins-guide-RNA ribonucleoprotein complexes (cas9-RNP) and Lipofectamine™ CRISPRMAX™ Cas9 Transfection Reagent according to the manufacturer’s instructions. Briefly, Cas9-RNP was prepared by incubating Cas9 protein (1.25 µL) with gRNA (240 ng) at a molar ratio of 5:1 at 25 °C for 10 min immediately prior to lipofection. Meanwhile, 23.5 µL Opti-MEM medium was added to a separate sterile Eppendorf tube, followed by the addition of 1.5 µL of Lipofectamine CRISPRMAX^®^. After brief vortexing, the Lipofectamine CRISPRMAX^®^ solution was incubated at room temperature for approximately 1 min. After incubation, the Lipofectamine CRISPRMAX^®^ solution was then added to the Cas9 solution. Upon mixing, the sample was incubated at room temperature for 10–15 min to form Cas9 and Lipofectamine CRISPRMAX^®^ complexes and then were added to the cells. Cells were transfected for 48 h with 50 µL CRISPR-Cas9/gRNA/transfection reagent complex per well and then were collected by trypsinization. To check for genome modification efficiency, GeneArt^®^ Genomic Cleavage Detection Kit was used to perform cell lysis, genomic PCR, and detection of cleavage, according to the manufacturer’s instructions. This method detects locus-specific DNA double-strand breaks formation by direct polymerase chain reaction (PCR) amplification and endonuclease activity that specifically cuts at hetero-duplex mismatches. HPRT1 locus edition was used as a positive control. Briefly, genomic DNA was purified using the PureLink^®^ Genomic DNA kit as described in the manual provided by the manufacturer from previously collected cells and amplified as described above. The resulting PCR product (2 µL) was mixed with the Detection Enzyme according to provided information. The digested product was analyzed with a 2% E-gel EX agarose. Syngene Genetools’ ultraviolet imaging system was used to visualize and image the gel. Syngene Genetools 4.03.05.0 (Synoptics Ltd., Cambridge, United Kingdom) software was used to quantify the optical density of DNA strands. The percentage of cleavage was quantified using the formula given by the manufacturer (% indel = 1 − ((1 − fraction cleaved)1/2).

After cleavage confirmation, transfection was again performed using validated gRNA, as described, and the selection of Caco-2 clones was carried out by limited dilution of KO cells in a 96-well plate. The isolated cell was cultivated and expanded into a Petri dish to reach 80% confluence. Cells were collected, and the genomic DNA was purified as described above. We then performed a PCR amplification with AmpliTaq^®^ Gold 360 Master Mix in the presence of the corresponding forward and reverse primers (see Appendix A) following the manufacturer’s instructions, with 2 µL of gDNA 20 µL reaction. The annealing temperature was 72 °C for all samples. One test reaction (edited gDNA) and one control reaction (unedited gDNA) were run for each gRNA target editing site. Samples were run on an agarose gel to confirm a product of the appropriate size. A clean PCR product was submitted for Sanger sequencing to GenoScreen (Lilles, France) using forward and reverse primers. Sanger results were aligned and analyzed using the Chromas 2.6.6 Software (Technelysium Pty Ltd., South Brisbane, Australia). Sanger results were then analyzed by the CRISPR Performance Analysis Tool by Synthego (Synthego Performance Analysis, ICE Analysis, 2019, v3.0) to assess the knock-out performance.

### 2.6. Off-Target Analysis

Potential unwanted mutations on off-target sites were determined in silico using an algorithm designed to identify candidate gRNA target sites after selecting the required PAM sequence (available at https://cm.jefferson.edu/Off-Spotter/, accessed on 22 July 2022 ) [16]. From the obtained results, we extracted 30 potential sites targeted by *MTTP*-gRNA and 15 potential sites targeted by *SAR1B*-gRNA. We then determined the off-target probability for each site (Off-Target Scoring) using the DeepCRISPR-1.0 R package [17]. This probability, or score, corresponds to cleavage frequency on the off-target site. Data analysis was performed using RStudio 2022.02.0 Build 443. Off-target sites with cleavage frequency below 10^−5^ were considered unlikely. Other off-target sites were amplified in each clone and sequenced by the Sanger method to check for mutations (Appendix A in Appendix A). Potential mutations were then spotted by aligning the clones’ sequences with the control sequences.

### 2.7. Western Blot Analysis

After being cultured in T25 flasks for 21 days (14 days post-confluence), cells were washed twice with PBS, scraped in 2 mL of PBS containing a protease inhibitor mix (1:200 dilution) and transferred into a sterile Eppendorf tube. Each sample was then vortexed for 15 min and sonicated for another 15 min while being maintained at 4 °C. The cell lysates were then centrifuged (680× *g*, 5 min, 4 °C). Supernatants were collected and transferred into new tubes. The protein content of the cell lysates was measured using a Pierce™ BCA Protein Assay Kit (kit 23225; Thermo Fisher Scientific). Equal amounts of proteins (50 µg) were separated by SDS-PAGE and transferred to the PVDF membrane. Membranes were incubated with a 1:500 dilution of specific primary antibody raised against Sar1b, MTP or β-Actin overnight at 4 °C. β-actin was used as a loading control for Western blot analysis. After the membranes were washed three times in Tris Buffered Saline Buffer with Tween 20 (TBST), the immunoblots were incubated with appropriate horseradish peroxidase-conjugated secondary antibodies (1:10,000) at room temperature for 1 h. Bands were visualized with enhanced chemiluminescence (ECL; Pierce by Thermo Fisher Scientific) and detected with ImageQuant^®^ LAS-4000 mini (GE Healthcare, Buc, France).

### 2.8. RNA Extraction and Quantitative RT-PCR

Cell lines were grown on filters as described above. RNA extraction and subsequent quantitative RT-PCR analyses were used to determine *MTTP* and *SAR1B* expression levels and the impact of *MTTP* and *SAR1B* silencing on the expression of other genes related to lipid metabolism in the different clones. Total RNA was extracted using a TRIzol reagent (Invitrogen™) according to the manufacturer’s instructions. One microgram of total RNA was reverse transcribed into cDNA using M-MLV Reverse Transcriptase (Invitrogen™), and amplification was achieved using Light Cycler^®^ 480 (Roche Molecular Systems, Rotkreuz, Switzerland). The 18S rRNA was used as the endogenous control in the comparative cycle threshold (CT) method. Data were analyzed using the LC480 software (Roche Diagnostics, Penzberg, Germany). The mRNA levels were calculated for each sample using the cycle threshold (Ct) and the ∆-CT method. The wild-type cells’ mRNA levels were assigned a value of 1, and the clone mRNA levels were expressed as relative expressions to the control. The primer sequences are reported in Appendix A Appendix A.

### 2.9. Optical Microscopy

*FBS-loaded cells imaging*: KO and control cells were grown on 24-well plates for 21 days. One day prior to the experiment, the medium used in apical and basolateral chambers was changed for a 16% FBS-complete medium. On the day of the experiment, cell monolayers were washed twice with 0.5 mL of PBS and stained with Oil Red O (Sigma-Aldrich). Oil Red O solution was prepared from an isopropanol solution saturated with Oil Red O diluted in water (2:3 *v*:*v*). The cells were then fixed in the culture wells by incubation for 15 min with a 10% aqueous solution of acetic acid and 10% methanol. All images were collected with a Nikon Eclipse TS100 (Nikon; Tokyo, Japan) light microscope equipped with a Nikon ELWD 0.3/OD75 C-A-O microscope lens. Images were acquired with the Nikon NIS Elements software (NIS-Elements F 2.3, SP3) at three different magnifications (20×, 100×, 400×). Images were analyzed using the freely available imaging software ImageJ (ImageJ, 1.53k; National Institutes of Health, USA).

*Mixed micelle-loaded cells imaging for intracellular lipid quantification*: KO and control cells were grown on cell culture filter inserts as described above. Twenty-four h prior to each experiment, both apical and basolateral medium were replaced by FBS-free medium. Cell monolayers were then incubated with synthetic mixed micelles, prepared as described by Reboul et al. [18], or with serum-free medium for 16 h. On the day of the experiment, cell monolayers were washed twice with 0.5 mL of PBS, fixed with paraformaldehyde, and stained with Oil Red O solution extemporaneously prepared from a saturated isopropanol solution of Oil Red O diluted in water (2:3 *v*/*v*), and finally mounted on microscope slides. All images were collected under light microscopy (Olympus BX63, 60 × 10). Intracellular lipid droplets were quantified as described by Vincent et al. [19] after acquiring 20 images per slide by standard light microscopy. Oil Red O-stained cell images were thresholded for the lipid droplet signal using the freely available imaging software ImageJ. The condition without micelles was used as a basal condition for each clone. The ratio of the augmented Oil Red O staining area between the condition in the presence of micelles and the condition in the absence of micelles was calculated. The number and average droplet size were determined by counting the number of droplets observed in the field of view and expressed as the ratio between the condition in the presence of micelles and the condition in the absence of micelles. All ratios were compared to the ratio calculated for control cells.

### 2.10. Assessment of Triglyceride and Cholesterol Basolateral Secretion

Cell lines were grown on filters as described above. Twenty-four h before the experiment, both apical and basolateral medium were replaced by FBS-free medium. On the day of the experiment, cell monolayers were washed twice with 0.5 mL of PBS. The apical side of the cell monolayers then received synthetic mixed micelles [18] made in DMEM without phenol red, whereas the basolateral side received a serum-free complete medium made in DMEM without phenol red. After 16 h, the basolateral medium was harvested. Aliquots were extemporaneously taken to evaluate secreted particle size and zeta potential with a Zetasizer Nano ZS (Malvern Instruments, Palaiseau, France). To this aim, 10 measurements per sample were carried out at 25 °C. The remaining samples were stored at −80 °C before cholesterol and triglyceride content was analyzed using fluorimetric quantification assays Abcam Picoprobe^®^ ab178780 (Abcam, Amsterdam, The Netherlands) and Amplite^®^ cholesterol quantification kit 40006 (AAT Bioquest, Euromedex, Souffelweyersheim, France), respectively, according to the manufacturer’s instructions. The samples were read at Ex/Em = 535/587 nm and Ex/Em = 540/590 nm using an Ensight Multimode Plate Reader instrument (PerkinElmer, Villebon-sur-Yvette, France). Experiments were repeated at least twice.

### 2.11. Assessment of Vitamin E (α-Tocopherol) Basolateral Secretion

Mixed micelles enriched with 100 µM vitamin E (α-tocopherol, TAC or TPGS) were prepared as previously described [18]. Cell lines were grown on filters as described above. Twenty-four h before the experiment, both apical and basolateral medium were replaced by FBS-free medium. On the day of the experiment, cell monolayers were washed twice with 0.5 mL of PBS. The apical side of the cell monolayers then received mixed micelles, whereas the basolateral side received a serum-free complete medium. After 6 h, media from each side of the cell monolayer were harvested. Cells were carefully rinsed twice with 0.5 mL ice-cold PBS before being collected in 0.560 mL PBS. All samples were stored at −80 °C before further analyses. Experiments were repeated at least twice.

### 2.12. Vitamin E Extraction and HPLC Analysis

Alpha-tocopherol and TAC were extracted from 1000, 500 or 250 µL of aqueous samples using the following method. Distilled water was added to sample volumes below 500 µL to reach a final volume of 500 µL. Gamma-tocopherol, which was used as an internal standard, was added to the samples in 500 µL of ethanol. The mixture was extracted twice with two volumes of hexane. The hexane phase obtained after centrifugation (1200× *g*, 10 min, 4 °C) was collected and evaporated to dryness under nitrogen, and the dried extracts were dissolved in 200 µL methanol. A volume of 50–200 µL was used for HPLC analysis. TPGS was indirectly assayed by quantifying α-tocopherol released after saponification by potassium hydroxide (KOH 5.5% + pyrogallol 1%) for 10 min at room temperature and by subtracting α-tocopherol determined before saponification as previously described [13].

Compounds were separated using a 250 × 4.6 mm, 5 µm Zorbax Eclipse XDB-C18 column (Agilent Technologies, Les Ulis, France) preceded by a guard column, maintained at a constant temperature (40 °C). Vitamin E and TAC analysis were performed with a 100% methanol mobile phase (flow rate = 1.5 mL·min^−1^). The HPLC system comprised a separation module, a photodiode array detector and a fluorescence detector (Ultimate 3000, Thermo Ficher Scientific). Compounds were detected at their maximum absorption wavelengths: 292 nm, and for fluorometric analysis, they were detected at 325 nm after light excitation at 292 nm. Vitamins were identified by retention time and absorption spectra coincident with authentic (>95% pure) standards. Quantification was performed using Chromeleon software (version 7.2; Thermo Fisher Scientific).

### 2.13. Statistical Analysis

Statistical tests were conducted using GraphPad Prism software version 8.4.3 (GraphPad Software, San Diego, CA, USA). Data were expressed as means ± SEM and tested, along with residuals (residual plot, homoscedasticity plot and QQ plot), for normal distribution and equality of variances before statistical testing. When appropriate, if heteroscedasticity was detected in the residuals, data were log-transformed before further parametric analysis was performed. Each group of unpaired data was compared to the control group using the ANOVA test. Post-hoc Dunnett’s test was used to compare the means of the different groups. For all tests, the bilateral alpha risk was α = 0.05. A *p*-value < 0.05 was considered significant.

## 3. Results

### 3.1. Selection of Efficient gRNA for Silencing of MTTP or SAR1B by CRISPR/Cas9 in Caco-2/TC7 Cells

For each gene, we designed two gRNA (*MTTP*-gRNA1 and *MTTP*-gRNA12 and *SAR1B*-gRNA1 and *SAR1B*-gRNA2) as a first step, and we aimed to select the most efficient one. Thus, four preliminary transfections using either *MTTP*-gRNA1, *MTTP*-gRNA2, *SAR1B*-gRNA1 or *SAR1B*-gRNA2 were performed. We then compared the transfection efficiency using a specific endonuclease. This method detects loci where the gene-specific double-strand breaks occur by direct polymerase chain reaction (PCR) amplification and endonuclease activity that specifically cleaves DNA at hetero-duplex mismatches. The resultant smaller bands, analyzed by gel electrophoresis, prove that mutated DNA could be targeted and cut by the Digestion Enzyme. Our results showed that only one gRNA led to mutated DNA for each gene. Transfections were further performed with *MTTP*-gRNA2 targeting the third exon of the *MTTP* gene and *SAR1B*-gRNA1 targeting the sixth exon of the *SAR1B* gene (Figure 1).

### 3.2. MTP and Sar1b Proteins Are Not Expressed in KO Clones

We obtained 47 living clones transfected with the *MTTP*-gRNA and 57 living clones transfected with the *SAR1B*-gRNA. DNA sequencing shows two kinds of results: (1) fully determined sequences with no mutation or (2) several overlapping sequences (Figure 2a,b). The presence of several overlapping sequences is likely due to the fact that Caco-2’s modal chromosome number is 96, occurring at 16%, with polyploidy at 3.2%.

The putatively mutated clones were therefore selected for a wider characterization. At this step, 13 clones transfected with the *MTTP*-gRNA and 12 transfected with the *SAR1B*-gRNA displayed disrupted sequences. We then analyzed CRISPR edits using Sanger data with the “Inference of CRISPR Edits” (ICE) tool, which estimates the frequency of impaired genotypes given the cut site. This analysis confirmed the absence of non-mutated sequences for *MTTP*-KO1 and *MTTP*-KO2 and showed the presence of non-mutated sequences for *SAR1B*-KO1 and *SAR1B*-KO2 at a lower frequency than mutated sequences, confirming a good knock-out efficiency.

Western blotting was then performed on the selected clones to detect the effect of mutations introduced with the CRISPR/Cas9 gene editing system on the expression of proteins of interest (Figure 3a,b). No detectable amount of MTP was observed on two clones, and no detectable amount of Sar1b was observed on two other clones, indicating a successful *MTTP* or *SAR1B* gene knock-out. These four clones were selected for further work and named clones *MTTP*-KO1 and *MTTP*-KO2 and clone *SAR1B*-KO1 and *SAR1B*-KO2.

We then tested whether the CRISPR/Cas9 mediated mutations of the affected gene’s mRNA synthesis by RT-qPCR (Figure 4a,b). The two clones with mutations in the *MTTP* gene and no detectable protein showed a significant decrease (*p* < 0.05) in gene expression with a mean *MTTP* expression level of 0.95 ± 0.15 in wild-type cells versus 0.42 ± 0.08 (*MTTP*-KO1) and 0.11 ± 0.01 (*MTTP*-KO2) in KO clones (Figure 4a). *SAR1B* gene expression was decreased in the two KO Caco-2/TC7 cell clones selected as compared with control cells (*p* < 0.05). Indeed, the mean expression level of the *SAR1B* gene was 1.90 ± 0.06 in control cells versus 0.21 ± 0.02 (*SAR1B*-KO1) and 0.09 ± 0.01 (*SAR1B*-KO2) in KO clones (Figure 4b).

### 3.3. No Off-Target Has Been Identified

In silico analyses suggested that (i) five genes could be the sites of off-target mutations in *MTTP*-KO clones: *BRPF1*, *IL34*, *MAN1A1*, *MCF2L* and *TNBX*, and (ii) two genes could be the sites of off-target mutations in *SAR1B*-KO clones: *CSAD* and *SCUBE3* (Appendix A).

Sequence alignment showed 100% alignment between clones’ sequences and control sequences, meaning that no sequence modifications were observed in potential off-target sites.

### 3.4. Expression of Other Genes Is Modulated in Knock-Out Clones Compared with Control Cells

Figure 5 shows the expressions of genes involved in intestinal lipid metabolism and oxidative stress management in the different clones.

*NPC1L1* expression was significantly decreased (*p* < 0.0001) in all KO clones with a mean of 1.00 ± 0.08 for wild-type cells versus a mean of 0.29 ± 0.03 (*MTTP*-KO1, *p* < 0.0001), 0.10 ± 0.01 (*MTTP*-KO2, *p* < 0.0001), 0.14 ± 0.02 (*SAR1B*-KO1, *p* < 0.0001) and 0.07 ± 0.01 (*SAR1B*-KO2, *p* < 0.0001).

*DGAT1* expression was also slightly but significantly (*p* < 0.05) decreased in all KO clones with a mean of 1.00 ± 0.04 for wild-type cells versus a mean of 0.49 ± 0.01 (*MTTP*-KO1, *p* < 0.0001), 0.55 ± 0.02 (*MTTP*-KO2, *p* < 0.0001), 0.78 ± 0.08 (*SAR1B*-KO1, *p* = 0.0084) and 0.48 ± 0.09 (*SAR1B*-KO2, *p* < 0.0001).

All *MTTP*-KO clones displayed significantly increased expression of *PLIN2* (1.00 ± 0.11 for the control group vs. 2.81 ± 0.33 and 2.28 ± 0.20 for the KO clones, *p* < 0.0001) as well as a significant decrease in *ABCA1* expression (1.00 ± 0.07 for the control group vs. 0.58 ± 0.02 and 0.53 ± 0.4 for the KO groups, *p* < 0.0001).

All KO clones presented a significant decrease of *ABCG8* expression (1.00 ± 0.14 for the control group vs. 0.23 ± 0.05 and 0.10 ± 0.01 for the *MTTP*-KO clones and 0.55 ± 0.16 and 0.07 ± 0.01 for the *SAR1B*-KO, *p* < 0.005). Both *MTTP*-KO clones also showed a significant decrease in *APOB* expression (1.00 ± 0.07 for the control group vs. 0.47 ± 0.06, *p* < 0.0001 and 0.68 ± 0.09, *p* = 0.0015 for *MTTP*-KO1 and *MTTP*-KO2, respectively).

Finally, *SAR1B*-KO1 displayed an increased expression of both FAPB2 and MGAT2, while *SAR1B*-KO2 presented a significant decrease in these genes’ expression. Regarding *FATP4*, *ACAT2*, *DGAT2*, *PGP*, *CAT* and *SOD*, only one or two clones showed small significant differences in expression compared with wild-type cells, with no particular pattern emerging.

### 3.5. Lipids Accumulate in Knock-Out Clones Compared with Control Cells

Oil Red O staining in cells incubated for 24 h with 16% FBS-complete medium revealed a cytoplasm filled with lipid droplets (Figure 6a–e). These lipid droplets appeared significantly larger and/or more numerous in KO clones than in control cells.

To confirm these observations, we compared and quantified intracellular lipid accumulation in cells after mixed micelle delivery. For the control cells, the ratio of the augmented Oil Red O staining area between the condition in the presence of micelles and the condition in the absence of micelles was 2.2. For the *MTTP*-KO1 clone, this ratio did not significantly differ from that of control cells. For *MTTP*-KO2, *SAR1B*-KO1 and *SAR1B*-KO2 clones, a significantly greater lipid accumulation was observed with ratios of 4.8 (+122 ± 18.4%), 4.1 (+89.2 ± 10.8%) and 3.6 (+67.9 ± 15.1%), respectively (*p* < 0.0001, Figure 7a). We observed a significant increase in the number of lipid droplets for *MTTP*-KO2 (+108.2 ± 26.4%, *p* < 0.0001) and *SAR1B*-KO1 clones (+80.4 ± 12.4%, *p* = 0.0004), but not for *MTTP*-KO1 and *SAR1B*-KO2 clones (Figure 7c). The results also indicate that *MTTP*-KO1, *MTTP*-KO2 and *SAR1B*-KO2 clones displayed a significant increase in cytoplasmic lipid droplet size compared with control cells (+24.7 ± 7.2% for *MTTP*-KO1, +23.6 ± 7, 6% for *MTTP*-KO2 and +34.5 ± 6.3% for *SAR1B*-KO2, respectively, *p* < 0.005; Figure 7b).

### 3.6. Cholesterol and Triglyceride Basolateral Secretion Is Decreased in Knock-Out Clones Compared with Control Cells

Compared with control cells, significant decreases in triglyceride and cholesterol secretion were observed after mixed micelles were delivered to KO cells. The mean reduction in cholesterol was −56.5 ± 7.8% for *MTTP*-KO1 (*p* < 0.0001), −35.3 ± 4.4% for *MTTP*-KO2 (*p* = 0.00025), −60.6 ± 3.5% for *SAR1B*-KO1 (*p* < 0.0001) and −57.8 ± 4.8% for *SAR1B*-KO2 (*p* < 0.0001) (Figure 8a). The mean reduction in triglycerides was −57.0 ± 2.6% for *MTTP*-KO1 (*p* < 0.0001), −74.0 ± 4.6% for *MTTP*-KO2 (*p* < 0.0001), −83.9 ± 1.6% for *SAR1B*-KO1 (*p* < 0.0001) and −75.4 ± 2.2% for *SAR1B*-KO2 (*p* < 0.0001) (Figure 8b). No impact on cholesterol and triglyceride uptake was observed.

Secreted particles were characterized in terms of size and zeta potential. Results are reported in Table 1. No significant difference in particle size was observed, even though a wider range of sizes was observed for KO clones.

### 3.7. Vitamin E Basolateral Secretion Is Decreased in Knock-Out Clones, Depending on Vitamin E’s Form

Results showed a significant decrease in α-tocopherol basolateral secretion with a mean reduction of −85 ± 7% for *MTTP*-KO1 (*p* < 0.0001), −77 ± 12% for *MTTP*-KO2 (*p* = 0.0002), −42 ± 4% for *SAR1B*-KO1 (*p* = 0.0338) and −57 ± 9% for *SAR1B*-KO2 (*p* = 0.0034) compared with control cells (Figure 9a). Although secretion was significantly impaired, no significant cytoplasmic difference was observed in vitamin E cytoplasmic levels in KO clones compared to wild-type cells.

We observed a significant decrease in TAC secretion in *MTTP*-KO clones with a mean decrease of −60 ± 3% in *MTTP*-KO1 (*p* = 0.027), −66 ± 13% in *MTTP*-KO2 (*p* = 0.046) compared with control cells (Figure 9b). The secretion of free α-tocopherol originating from TAC was impaired to a greater extent in these clones with a decrease of −76 ± 5% in *MTTP*-KO2 (*p* < 0.0001), and no detectable amount (*p* < 0.0001) in *MTTP*-KO1 basolateral medium compared with the control. Regarding *SAR1B*-KO clones, TAC basolateral was decreased, but the difference remained non-significant: −40 ± 15% (*p* = 0.132) and −25 ± 10% (*p* = 0.598) for *SAR1B*-KO1 and *SAR1B*-KO2, respectively. For these two clones, we also observed a significant decrease in free α-tocopherol secretion compared with wild-type cells: −57 ± 15% (*p* = 0.0007) and −71 ± 10% (*p* < 0.0001) for *SAR1B*-KO1 and *SAR1B*-KO2, respectively (Figure 9b).

Finally, we observed no significant difference in TPGS basolateral secretion between control cells and all KO clones after incubation of TPGS diluted in mixed micelles, with a very low secretion for both control and KO cells. The secretion of free α-tocopherol originating from TPGS was also notably low (4.8 ± 1.1 pmol/well for wild-type cells vs. 3.5 ± 0.4 pmol/well for *MTTP*-KO1, 2.7 ± 1.3 pmol/well for *MTTP*-KO2, 1.6 ± 1.0 pmol/well for *SAR1B*-KO1 and no detectable amount for *SAR1B*-KO2), resulting in non-significant differences as well (Figure 9c).

Of note, esterified tocopherol secretion intensity depended on vitamin E source: when given TAC, wild-type Caco-2/TC7 secreted 388 ± 49 pmol/well of TAC, but when given TPGS wild-type Caco-2/TC7 secreted 12.6 ± 6.7 pmol/well of esterified tocopherol after 6 h.

## 4. Discussion

The objective of this study was to evaluate whether Caco-2/TC7 cells knocked out for *MTTP* or *SAR1B* genes could constitute relevant models for enterocytes of patients with Monogenic Hypobetalipoproteinemia Disorders. The Caco-2/TC7 sub-clone has shown its ability to absorb dietary lipids [20] and to secrete TAG-rich particles [21], which makes it a relevant cell line to develop a model for studying the small intestines of these patients. The CRISPR/Cas9 technique is a relatively simple genome editing system based on the nuclease Cas9 and a single gRNA that has been established in many organisms, from bacteria to higher eukaryotes [22]. Here, we successfully knocked out *MTTP* in two cell lines (named *MTTP*-KO1 and *MTTP*-KO2) and *SAR1B* in two cell lines (named *SAR1B*-KO1 and *SAR1B*-KO2), as confirmed by DNA sequencing and Western blotting (WB) showing mutated DNA sequences and lack of proteins. Gene expression, assessed by qPCR, revealed a low expression of either *MTTP* or *SAR1B* in the respective KO clones.

The four KO clones were characterized by light microscopy after Oil red O staining (a lipid-soluble dye that stains neutral lipids, including cholesteryl esters). The microscopy experiment with cells loaded with 16% FBS-complete medium showed a significant increase in cytoplasmic lipid droplets number and size in *MTTP*-KO1, *SAR1B*-KO1 and *SAR1B*-KO2 compared with the control, in accordance with histological abnormalities observed in intestinal biopsies of patients [9]. Intracellular lipid quantification determined under light microscopy on cells loaded with mixed micelles for 16 h confirmed the greater lipid accumulation in all KO clones. However, each clone displayed a different type of response to micellar delivery. Clone *MTTP*-KO1 had larger but slightly fewer cytoplasmic droplets, which does not increase the Oil red O staining area compared to control cells, whereas clone *MTTP*-KO2 had both larger and more numerous lipid droplets, which explains the larger increase in staining area compared with the other clones. The *SAR1B*-KO1 clone has lipid droplets comparable in size to control cells but much more numerous, while the *SAR1B*-KO2 clone has lipid droplets comparable in number to control cells but of greater size.

We then studied the impact of knocking out *MTTP* or *SAR1B* on the expression of different genes involved in lipid metabolism [19] and oxidative stress management. We explored the expressions of genes involved in fatty acid transport (*FATP4*, *FABP2*), intracellular lipid re-synthesis (*ACAT2*, *DGAT1*, *DGAT2*, *MGAT2*), lipid droplet (*PLIN2*) or chylomicron (*APOB*) formation, cholesterol metabolism (*ABCA1*, *ABCG8*, *NPC1L1*, *SCARB1*, *PGP*) and oxidative stress management (*CAT*, *SOD1*). Our results highlight a decrease in *NPC1L1* expression, a gene expressed in the small intestine and required for intestinal cholesterol absorption [23], in all clones. Studies from Xie et al. [24] and Iqbal et al. [25] also reported a subsequent decrease in *NPC1L1* expression in isolated enterocytes obtained from mice with conditional intestine-specific *MTTP* deficiency. The *NPC1L1* gene may be downregulated in response to an increase in cellular lipid amounts. Indeed, an increased amount of cholesterol or its esters, known to be unstable in oxidizing conditions, might lead to an elevated risk of oxysterol formation [26]. As a result, the Liver X Receptors (LXRs), which act as oxysterol sensors [27], would induce the transcription of genes that protect cells from cholesterol overload [28]. *NPC1L1* was recently reported to be downregulated by LXR activation in both mice and a human enterocyte cell line [29]. The observed *NPC1L1* downregulation may thus be a consequence of LXR activation. However, this hypothesis is inconsistent with the marked *ABCA1* downregulation in three clones, as LXR is also a positive regulator of *ABCA1* [30]. However, *ABCA1* down-regulation in the two *MTTP*-KO clones is also in agreement with expression levels reported by Xie et al. in mice with conditional intestine-specific *MTTP* deficiency [24]. Interestingly, the expressions of *ABCG8* (coding a transporter involved in non-esterified sterol intestinal excretion [31]) and of *APOB* (coding for the main structural protein of chylomicrons) were significantly decreased in both *MTTP*-KO clones, still in agreement with the study from Xie et al. [24]. Intestinal *ABCG8* downregulation was also observed by Iqbal et al. in their in vivo model, but the authors did not evaluate *APOB* levels [25]. Overall, these results suggest that these two clones are valid models of *MTTP* deficiency.

The increased expression of *PLIN2* in *MTTP*-KO clones is an interesting result. PLIN2 (Perilipin 2/adipophilin) is the major and ubiquitous lipid droplet coat protein and plays a major role in the biogenesis, stabilization and degradation of cytoplasmic lipid droplets. PLIN2 only exists bound to lipid droplets and is known to be a specific marker of lipid accumulation [32]. Bouchoux et al. showed in Caco-2/TC7 cells that PLIN2 labeling increased in lipid-loaded cells compared with control cells after a 24 h incubation with lipid micelles [33]. *PLIN2* expression was also higher in enterocytes from chronic high-fat challenged mice compared with chow-fed mice [34]. This result is also consistent with the greater lipid droplet size observed during microscopy experiments.

*SAR1B*-KO2 displayed downexpression of *ABCA1* and *MTTP* genes, which is consistent with a previous study on a Sar1b mutant mice model [35]. Conversely, *SAR1B*-KO1 showed no alteration in both *ABCA1* and *MTTP* expression. However, both *SAR1B*-KO clones showed a significant reduction of *ABCG8*, NPC1L1 and DGAT1 expressions, as observed in *MTTP*-KO clones. Moreover, *SAR1B*-KO1 displayed an increased expression of *FABP2*, encoding for an intestinal fatty acid-binding protein involved in the uptake, intracellular metabolism and transport of long-chain fatty acids [36], and of *MGAT2*, encoding for an intestinal protein that catalyzes the synthesis of diacylglycerols, a precursor of TAG [37], while *SAR1B*-KO2 displayed a decreased expression of these genes. Of note, *SAR1B*-KO2 showed a significant decrease in the expression of all tested genes. This variability may come from the selection of each clone from a single cell. Indeed, the Caco-2/TC7 line presents homogenous cells compared to other lines [38], but slight genomic and phenotypic differences still exist between cells from the same culture plate.

The observed lipid intracellular accumulation in both Caco-2/TC7 lacking MTP and Sar1b proteins was consistent with the cholesterol and triglycerides accumulation observed in enterocytes from intestine-specific *MTTP*-KO mice [24,25] and Sar1b mutant mice [35] compared with wild-type mice. We also observed altered basolateral triglycerides and cholesterol secretion that has been observed in both mouse models, with a significant decrease in plasma cholesterol levels in conditional intestine-specific *MTTP*-deficiency mice [24,25] and Sar1b mutant mice [35].

We then characterized the secretion of α-tocopherol, TAC and TPGS by wild-type Caco-2 cells compared to KO clones. Alpha-tocopherol is one of the main dietary forms of vitamin E. In clinical nutrition, vitamin E deficiency is usually treated with oral α-tocopherol-acetate (TAC) to protect the phenol group against oxidation [13,39]. Due to the poor bioavailability of TAC and because of the interest in treating patients with Monogenic Hypobetalipoproteinemia Disorders with TPGS, a drinkable form of tocopherol, we characterized and compared the uptake and secretion of α-tocopherol, TAC and TPGS in the different clones. Our main results are: (1) free α-tocopherol secretion was significantly decreased in all KO clones and to a greater extent in *MTTP*-KO clones compared with *SAR1B*-KO clones; (2) TAC secretion was mildly but significantly decreased in *MTTP*-KO clones but not in *SAR1B*-KO clones; (3) TPGS secretion, which was much lower than those of TAC and α-tocopherol, was not modified between wild-type and KO cells, probably due to the negligible amounts secreted. The fact that we observed a lower α-tocopherol secretion in *MTTP*-KO clones than in *SAR1B*-KO clones is consistent with clinical observations, as FHBL-SD1 patients present more severe vitamin E deficiency than FHBL-SD3 patients [40]. Our results are also consistent with a previous cell-based study designed to characterize the absorption of free tocopherol, TAC and TPGS [13]. This previous study also reported a basolateral secretion of TAC in its intact form (202 pmol/well after 24 h of incubation). Both studies showed a greater secretion of tocopherol when micellar tocopherol was in its free form than when it was in its esterified form (TAC). TAC secretion was modestly impacted in our KO clones, while the secretion of tocopherol derived from TAC hydrolysis was significantly reduced. This suggests that free tocopherol secretion is more dependent on intracellular lipid metabolism than its esterified form, likely because TAC secretion is mediated by an alternative (and relatively inefficient) pathway. Finally, TPGS secretion was not significantly modified between control cells and KO clones, probably due to the extremely small amount of TPGS secreted, according to previous observations [13]. These findings are in accordance with a clinical trial [41], which reported that TAC was better absorbed than tocofersolan-TPGS in healthy adult volunteers after a single oral dose. The authors also observed that supplementation with tocofersolan-TPGS had a minimal effect on plasma α-tocopherol in normal individuals.

Although our data suggest that our KO clones are a relevant model for Monogenic Hypobetalipoproteinemia Disorders, our clones present some limits. Firstly, limitations are linked to the CRISPR/Cas9 technique itself. To date, 75 causal mutations of FHBL-SD1 have been reported throughout the *MTTP* gene [42], and 16 different pathogenic variants of *SAR1B* have been reported [43] in different introns and exons. Here, we could only generate one mutation type in one specific exon of each gene, making it difficult to reproduce the high variability of genotypes and phenotypes found in the patients. Secondly, CRISPR/Cas9 may have induced unwanted off-target mutations that could not be detected with our in silico analyses. However, the effects are expected to be reduced because we used a lipid nanoparticle delivery system and appropriate target sites [44]. Thirdly, although disturbed intestinal absorption is a key element in these pathologies, one should not forget the potential role of the liver and others tissues that are also affected by the mutations, which may interfere with the metabolism and distribution of lipids and vitamin E.

## 5. Conclusions

Overall, we generated enterocyte models to study Monogenic Hypobetalipoproteinemia Disorders. The selected clones showed (1) lipid droplet and cholesterol cellular accumulation; (2) downregulation of *NPC1L1*, *ABCG8*, *ABCA1* and *APOB* in *MTTP*-KO clones and downregulation of *ABCA1* and *APOB* in one *SAR1B*-KO clone in accordance with in vivo data obtained from conditional *MTTP*-KO mice and *SAR1B*-KO mice, respectively; (3) a reduced secretory capacity of cholesterol and vitamin E, as observed in FHBL-SD1 and FHBL-SD3 patients. Our KO clones thereby represent interesting tools to further experiment with new therapeutic strategies to improve lipid and vitamin E absorption in patients.

## Figures and Tables

**Figure 1 nutrients-15-00505-f001:**
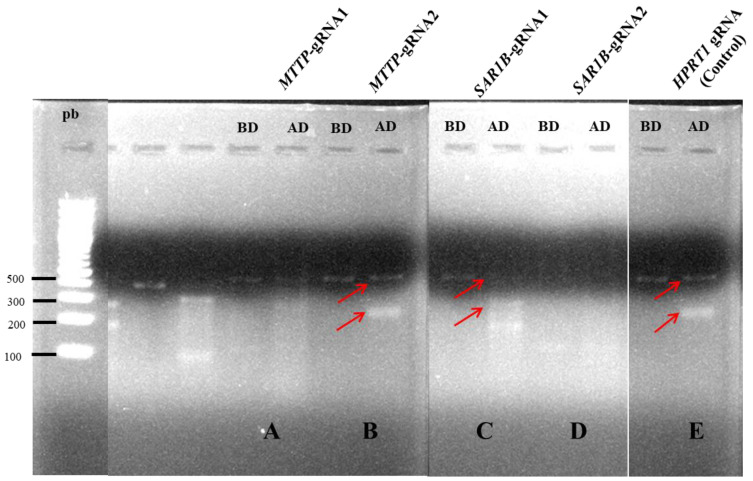
Confirmation of gRNA efficiency later used for transfection and silencing of *MTTP* or *SAR1B* by CRISPR/Cas9. Amplification products detected on 1% E-gel EX agarose of genomic DNA extracted from Caco-2/TC7 cell line transfected with (**A**) gRNA1 targeting *MTTP*, (**B**) gRNA2 targeting *MTTP*, (**C**) gRNA1 targeting *SAR1B*, (**D**) gRNA2 targeting *SAR1B* and (**E**) gRNA targeting HPRT1, used as a control gene, before (**B**,**D**) the enzymatic digestion efficiency test and after (**A**,**D**) the enzymatic digestion efficiency test. This method detects locus-specific DNA double-strand breaks formation by direct polymerase chain reaction (PCR) amplification and endonuclease activity that specifically cuts at hetero-duplex mismatches. Only mutated DNA could be targeted and cut by the Digestion Enzyme, as evidenced by the presence of two bands, indicated by red arrows, with sizes congruent with the distance between Cas9 cutting sites.

**Figure 2 nutrients-15-00505-f002:**
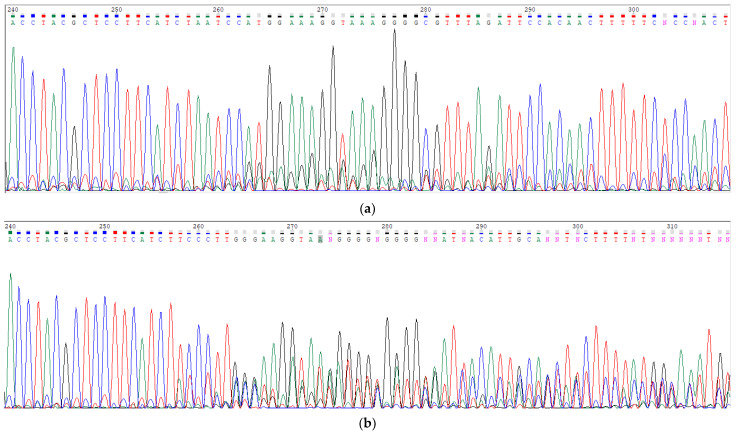
Example of DNA sequences of the targeted region in the *MTTP* gene, obtained by the Sanger sequencing method. Cells were cultured in Petri dishes to reach confluence, and DNA was extracted from each transgenic clone and wild-type Caco-2/TC7 cell (WT), which was used as a control. (**a**) The targeted DNA sequence of wild-type cells in the *MTTP* gene shows a fully determined unique sequence. (**b**) The targeted DNA sequence in the *MTTP* gene in clone *MTTP*-KO1 shows overlapping disrupted sequences beginning at nucleotide 258.

**Figure 3 nutrients-15-00505-f003:**
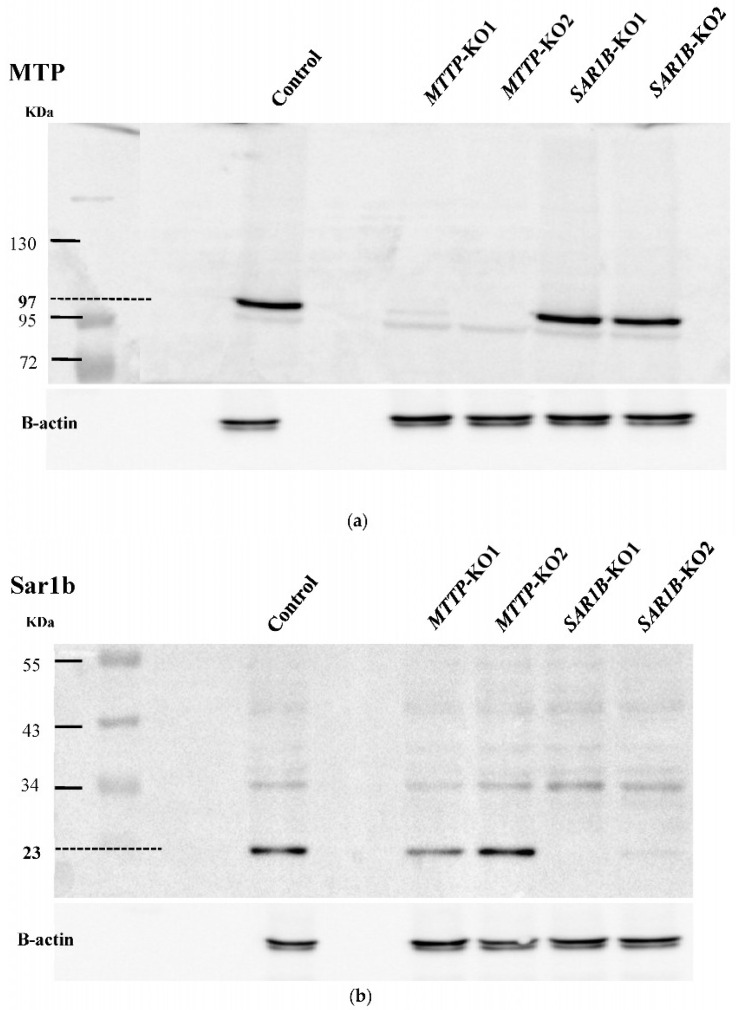
Knocking-out confirmation of *MTTP* or *SAR1B* by SDS-PAGE (4–10%) and Western blot analysis. Cells were cultured in T25 flasks for 21 days. Total protein was extracted from each transgenic clone and wild-type Caco-2/TC7 cells (WT), which were used as a positive control. Fifty μg of extracts were resolved by 10% SDS/PAGE, and the blot was stained with monoclonal antibodies raised against MTP or Sar1b and β-actin to ensure equal lane loading. (**a**) Analysis of MTP levels in two clones transfected with *MTTP*-gRNA (first two bands) and two clones transfected with *SAR1B*-gRNA (last two bands), revealing no 97 kDa band for the two first clones while being detected in wild-type cells (positive control) and in clones transfected with *SAR1B*-gRNA, highlighting two clones without protein, named *MTTP*-KO1 and *MTTP*-KO2. (**b**) Analysis of Sar1b levels in 2 clones transfected with *MTTP*-gRNA (first two bands) and two clones transfected with *SAR1B*-gRNA (last two bands), revealing no 23 kDa band for the two last clones while being detected in wild-type cells (positive control) and in clones transfected with *MTTP*-gRNA, highlighting two clones without protein named *SAR1B*-KO1 and *SAR1B*-KO2.

**Figure 4 nutrients-15-00505-f004:**
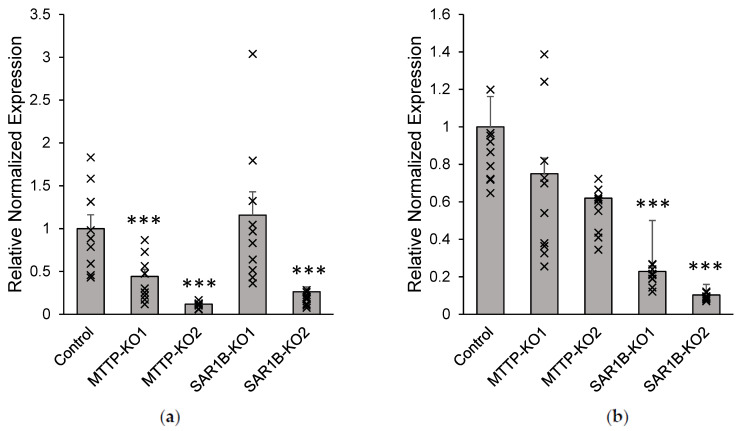
Gene expression measured by RT-qPCR in Caco-2/TC7 cells following CRISPR/Cas9 knock-out. Cells were cultured on inserts for 21 days. (**a**) *MTTP* and (**b**) *SAR1B* mRNA expression determined by RT-qPCR in two clones lacking MTP and two clones lacking Sar1b proteins (as seen on Western blots). Wild-type cells’ mRNA levels were assigned a value of 1. and *MTTP*- or *SAR1B*-KO mRNA levels are expressed as relative expressions to wild-type cells. Data are expressed as means ± SEM (*n* = 10). Asterisks indicate a statistically significant difference in gene expression levels between control and KO clones (*** *p* < 0.0001).

**Figure 5 nutrients-15-00505-f005:**
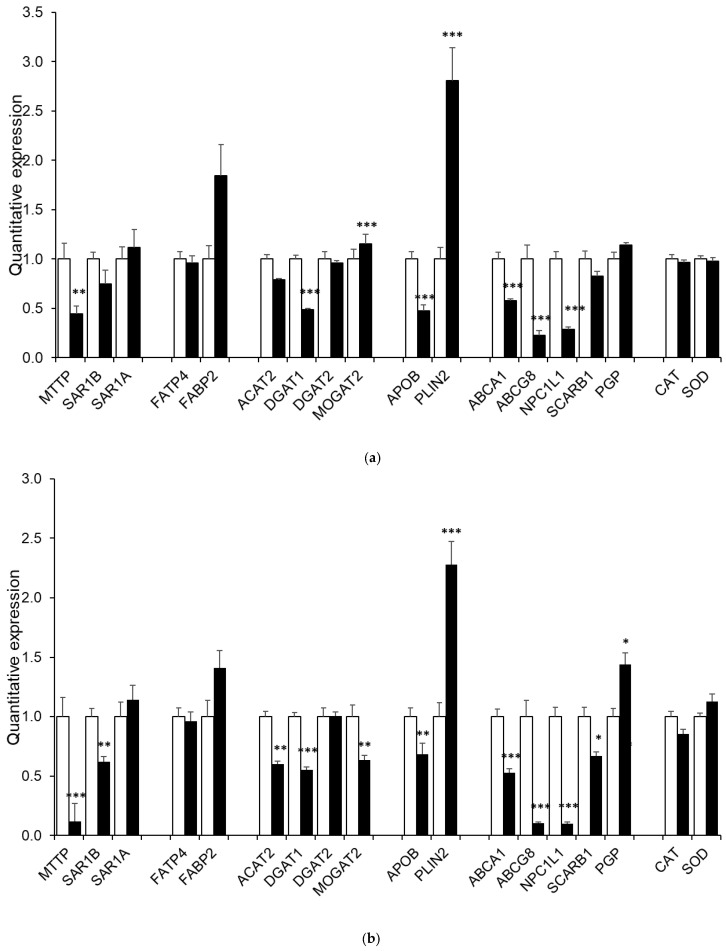
Characterization of clone gene expression. Cells were cultured on inserts for 21 days. We studied genes involved in fatty acid transport (*FATP4*, *FABP2*), intracellular lipid re-synthesis (*ACAT2*, *DGAT1*, *DGAT2*, *MGAT2*), lipid droplet (*PLIN2*) or chylomicron (*APOB*) formation, cholesterol metabolism (*ABCA1*, *ABCG8*, *NPC1L1*, *PGP*, *SCARB1*) and oxidative stress management (*CAT*, *SOD1*) in (**a**) *MTTP*-KO1, (**b**) *MTTP*-KO2, (**c**) *SAR1B*-KO1, (**d**) *SAR1B*-KO2. Wild-type cells’ mRNA levels were assigned a value of 1, and clones’ mRNA levels were expressed as relative expressions to wild-type cells. Gene expression was measured by RT-qPCR in KO and wild-type Caco-2/TC7 cells. Data are expressed as means ± SEM (*n* = 10). Asterisks indicate a statistically significant difference in gene expression levels between control and KO clones (* *p* < 0.05, ** *p* < 0.001, *** *p* < 0.0001). White bars represent control cells; black bars represent considered clones.

**Figure 6 nutrients-15-00505-f006:**
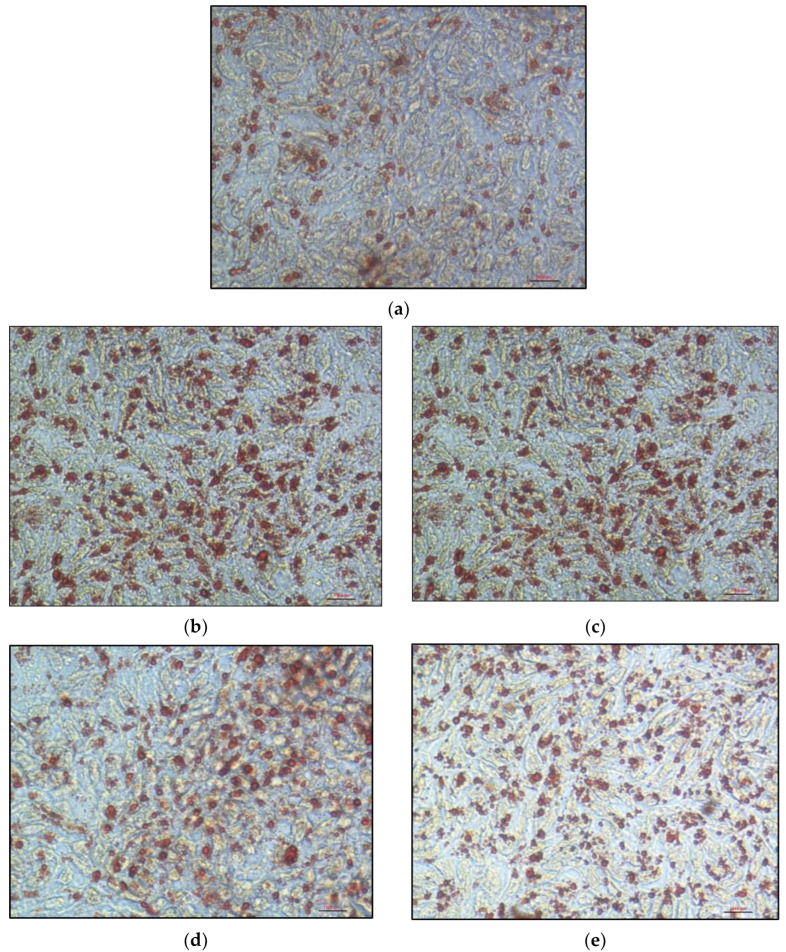
Representative images of cytoplasmic lipid droplets were observed under inverted microscopy images. Cells were cultured in 24-well plates for 21 days and received a medium containing 16% FBS (instead of 8% FBS) for 24 h. Oil Red O staining (×400 magnification) was performed in (**a**) wild-type Caco-2/TC7 cells, (**b**) clone *MTTP*-KO1, (**c**) clone *MTTP*-KO2, (**d**) clone *SAR1B*-KO1 and (**e**) clone *SAR1B*-KO2.

**Figure 7 nutrients-15-00505-f007:**
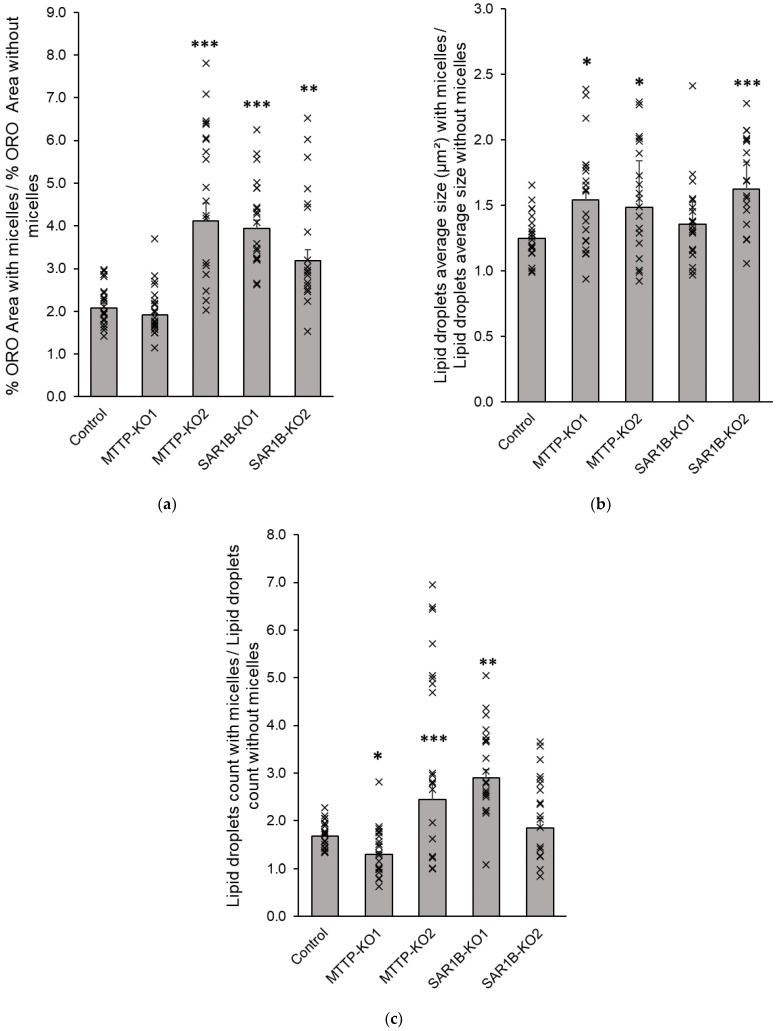
Lipid accumulation in KO clones and control cells. Cells were cultured on inserts for 21 days. Quantification of lipid accumulation was assessed by the ratio of increases in (**a**) red surface area (Oil red O staining), (**b**) lipid droplets count per field and (**c**) lipid droplets size between cells incubated without micelles and cells incubated with micelles for 16 h. Data are expressed as means ± SEM (*n* = 20). Asterisks indicate a statistically significant difference between control and KO clones (* *p* < 0.05, ** *p* < 0.001, *** *p* < 0.0001).

**Figure 8 nutrients-15-00505-f008:**
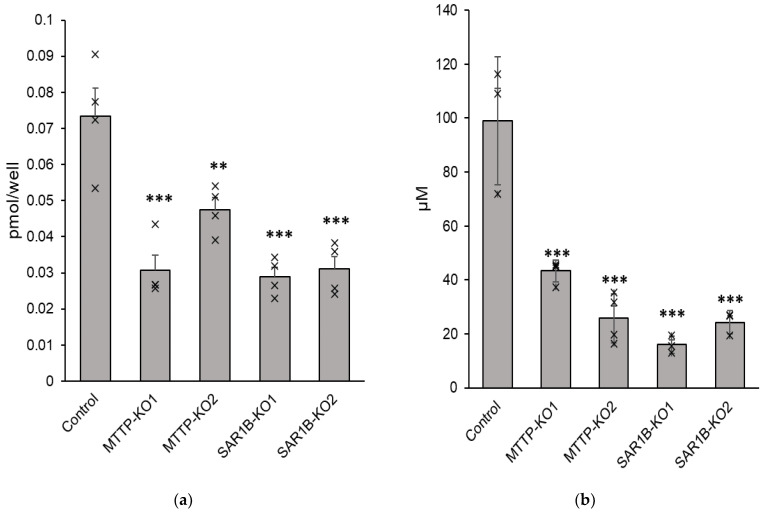
(**a**) Cholesterol and (**b**) triglyceride basolateral secretion in *MTTP* and *SAR1B* KO Caco-2/TC7 cells compared with wild-type Caco-2/TC7 cells. Cells were cultured on inserts for 21 days and were incubated with mixed micelles for 16 h. Triglyceride and cholesterol levels in the basolateral medium were quantified by fluorimetric assays. Data are expressed as means ± SEM (*n* = 4). Asterisks indicate a statistically significant difference in cholesterol concentrations between control and KO clones (** *p* < 0.001, *** *p* < 0.0001).

**Figure 9 nutrients-15-00505-f009:**
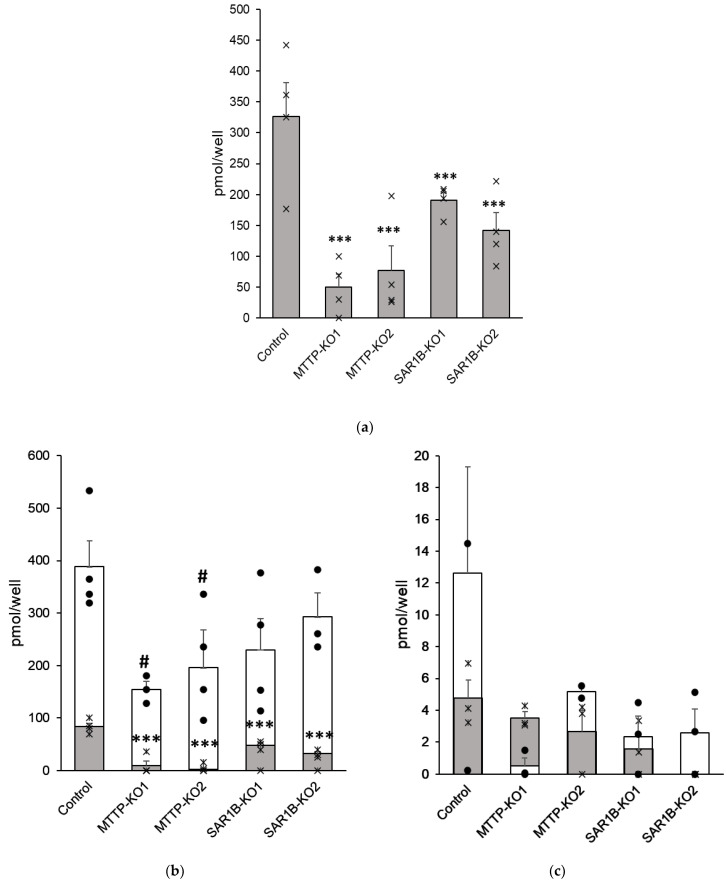
Vitamin E and its basolateral-derivative content after (**a**) free α-tocopherol, (**b**) TAC or (**c**) TPGS acute micellar delivery. Cells were cultured on inserts for 21 days and were incubated with mixed micelles containing α-tocopherol or TAC or TPGS for 6 h. Alpha-tocopherol and its derivative levels were measured by HPLC after liquid–liquid extraction with hexane. Data are expressed as means ± SEM (*n* = 4). For (**b**,**c**), grey bars with crosses represent free α-tocopherol and white bars with dots represent TAC (**b**) or TPGS (**c**). For all figures, asterisks indicate a statistically significant difference in TAC concentrations between control and KO clones (*** *p* < 0.0001) and a hash (#) indicates a statistically significant difference in TAC or TPGS concentrations between control and KO clones (# *p* < 0.05).

**Table 1 nutrients-15-00505-t001:** Physicochemical characteristics of particles secreted by Caco-2 cells.

Clones	Size (nm) Mean ± Sem (Population Range)	Zeta-Potential (mv) Mean ± Sem
Control	46 ± 2.7 (9–68)	−9.66 ± 1.0
*MTTP*-KO1	59 ± 14.0 (16–91)	−10.74 ± 0.92 ***
*MTTP*-KO2	61.9 ± 14.9 (18–122)	−9.56 ± 1.28
*SAR1B*-KO1	57.4 ± 2.5 (18–122)	−9.77 ± 1.0
*SAR1B*-KO2	61.5 ± 6.9 (16–122)	−8.93 ± 1.65 *

Cells were incubated for 16 h with mixed micelles. The basolateral medium was then harvested, and aliquots were extemporaneously taken to evaluate secreted particle size and zeta potential. Asterisks indicate a statistically significant difference between control and KO clones (* *p* < 0.05, *** *p* < 0.0001).

## Data Availability

Not applicable.

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
