# Peer review of "Validation of Knock-Out Caco-2 TC7 Cells as Models of Enterocytes of Patients with Familial Genetic Hypobetalipoproteinemias"

_nutrients, 2023, doi:10.3390/nu15030505_

Round 1
Reviewer 1 Report
The authors should include the DNA sequencing chromatograms of the KO lines and discuss the mutations found.
It also should be noted that CACO-2 cells have a modal chromosome number of ~96 which would influence efficiency of KO generation.
Author Response
The authors should include the DNA sequencing chromatograms of the KO lines and discuss the mutations found.
Example of chromatograms were added as Figure 2.
We also added the following text to explain how the mutated clones were selected:
Material and methods P5: “Sanger results were then analyzed by the CRISPR Performance Analysis Tool by Synthego (Synthego Performance Analysis, ICE Analysis. 2019. v3.0. Synthego) to assess the knock-out performance. »
Results P 9: “We then analyzed CRISPR edits using Sanger data with the “Inference of CRISPR Ed-its” (ICE) tool, which estimate the frequency of impaired genotypes, given the cut site. This analysis confirmed the absence of non-mutated sequences for MTTP-KO1 and MTTP-KO2 and the showed presence of non-mutated sequences for SAR1B-KO1 and SAR1B-KO2 at a lower frequency than mutated sequences, confirming a good knock-out efficiency.”
It also should be noted that CACO-2 cells have a modal chromosome number of ~96 which would influence efficiency of KO generation.
We specified it in the text P 9: “DNA sequencing shows two kinds of results: (1) fully determined sequences with no mutation or (2) several overlapping sequences (Figure 2A and 2B). The presence of several overlapping sequence is likely due to the fact that Caco2 modal chromosome number is 96, occurring at 16% with polyploidy at. 3.2%.”.
However, the analysis of the gene sequence as well as the Western Blots confirmed the fact that we selected mutated clones.
Reviewer 2 Report
The manuscript by Claire et al investigated knock-out cell models of FHBL- 25 SD1 and FHBL-SD3 using the CRISPR/Cas9 technique in Caco-2/TC7 cells. The current study overall is well-designed and the results presented to support their conclusion. I have some minor concerns described below:
1) Please revise all of the figures using professional graph-making software and make the graph style consistent throughout the manuscript. In some of the figures, the significant symbols are out of place.
2) Please rationale why 4.5g/L glucose level is considered as high glucose, and how this correlated with plasma glucose level in diabetic patients.
Author Response
The manuscript by Claire et al investigated knock-out cell models of FHBL- 25 SD1 and FHBL-SD3 using the CRISPR/Cas9 technique in Caco-2/TC7 cells. The current study overall is well-designed and the results presented to support their conclusion. I have some minor concerns described below:
1) Please revise all of the figures using professional graph-making software and make the graph style consistent throughout the manuscript. In some of the figures, the significant symbols are out of place.
The figures were initially made with Graphpad (Prism), which induced problem with edition.
Wi thus converted all the files into Excel files, which would lead to a better image quality in the World template provided by the Journal.
2) Please rationale why 4.5g/L glucose level is considered as high glucose, and how this correlated with plasma glucose level in diabetic patients.
This is actually the usual glucose level for Caco-2 cell culture.
The term “high glucose” has thus been removed and a reference to previous study we made with this media were added.